# Molecular Evolution and Genetic Variation of *G2-Like* Transcription Factor Genes in Wheat (*Triticum aestivum* L.)

**DOI:** 10.3390/genes14071341

**Published:** 2023-06-26

**Authors:** Ge Hu, Junchang Li, Xiang Wang, Yunfei Kang, Yongchun Li, Jishan Niu, Jun Yin

**Affiliations:** National Engineering Research Centre for Wheat/Henan Technology Innovation Centre of Wheat, Henan Agricultural University, Zhengzhou 450046, China

**Keywords:** wheat (*Triticum aestivum* L.), *G2-like*, expression profiles, abiotic stress

## Abstract

The GOLDEN2-LIKE (*G2-like*) gene family members provide significant contributions to the growth and development of plants. In this study, a total of 76 wheat *G2-like* gene family members (*TaG1–TaG76*) were detected in the wheat genome and were categorized into three groups (including six subgroups) based on the gene structure and protein motif analyses. These genes were unevenly distributed in 19 of 21 wheat chromosomes. A total of 63 segmental duplication pairs of *TaG2-like* genes were identified in the wheat genome. The expression levels of all the *TaG2-like* genes indicated that *TaG2-like* genes showed different expression patterns in various organs and tissues. Moreover, the transcriptions of *TaG2-like* genes were significantly affected under abiotic stress (cold, ABA, NaCl, and PEG). This study offered valuable insights into the functional characterization of *TaG2-like* genes in wheat.

## 1. Introduction

*G2-like* transcription factors belong to a subfamily of the GARP superfamily, which is defined as a type of transcription factor widely present in plants [1]. The G2-like protein was first identified to play a role in plant growth and development as transcription factors in maize (*Zea mays* L.) [2]. Most *G2-like* genes contain two conserved domains, including a Myb-DNA binding domain and a C-terminal domain [3].

Many studies have shown that *G2-like* gene family members play a crucial role in the plant’s development of their chloroplasts. In maize, the G2-like proteins were proven to be transcriptional regulators of the cell-type differentiation processes in the chloroplast development of C4 plant tissues [4]. The *G2-like* gene initially presented as a single gene in the original flowering plant, following which the replication of this gene has been observed in several species, which is related to the polymorphism of plant chloroplasts [5]. *G2-like* genes coordinate and synchronize the expression of nuclear photosynthetic genes in *Arabidopsis* [6]. In rice, AtGLK1 and AtGLK2 are expressed in photosynthetic tissues and regulate chloroplast development [7,8]. In tomatoes, the expression of the GOLDEN2-LIKE gene enhances fruit photosynthetic capacity, leading to an increased sugar content [9]. Additionally, the *G2-like* gene plays a significant role in the plant flowering time. OsPHL3, identified as one of the *G2-like* transcription factors, has been one of the proven to be involved in delaying the flowering time [10]. 

With the release of plant genomic sequences, the *G2-like* gene family members have been identified and analyzed in several plant species, including in *Arabidopsis thaliana* [7], maize [3], rice (*Oryza sativa* L.) [4], tomato [11], and tobacco [12]. These data significantly promoted the functional studies of the plant *G2-like* gene family. However, the *G2-like* gene family has not undergone comprehensive examinations in wheat. Wheat (*Triticum aestivum* L.) is an important food crop in the world. It could be observed that studies on the function of the *G2-like* gene family in wheat could provide a way to make the most of photosynthesis and increase wheat production in the future.

In this study, a thorough examination of all the wheat *G2-like* genes was conducted with the current wheat genome (RefSeq-v1.1) sequence data (IWGSC, 2018), including on the gene structure, phylogenetic relationship, synteny analysis, and the expression levels of the *G2-like* gene family members in the different tissues or organs of wheat. Meanwhile, the expression levels of *G2-like* genes under different abiotic stress conditions were systematically analyzed. The findings of this study will facilitate future functional investigations of the *G2-like* gene family in wheat.

## 2. Materials and Methods

### 2.1. Gene Identification

The genome version IWGSC refseqv1.1 obtained from http://plants.ensembl.org/ (accessed on 1 January 2023) was utilized to identify individuals of the *G2-like* family in wheat. *Arabidopsis* G2-like protein sequences (AtGLK1 and AtGLK2) [3] were utilized as search queries to compare with the wheat database with BLASTP. Additionally, the hidden Markov model (HMM) file associated with the domain (PF00249) was acquired from the Pfam database (https://www.ebi.ac.uk/interpro/search/sequence/, accessed on 1 January 2023). HMMER 3.3 (http://www.hmmer.org/, accessed 1 January 2023) [13] and was used to search the *G2-like* family members. Protein sequences with an e-value ≤ 1 × 10^−10^ were gathered as the output. To identify the *G2-like* genes, the longest transcript was considered, taking into account the possibility of multiple transcripts for each gene in the wheat genome.

All candidate genes were further examined using the CDD (https://www.ncbi.nlm.nih.gov/Structure/cdd/cdd.shtml, accessed on 1 January 2023) and SMART programs (http://smart.embl-heidelberg.de/smart/set_mode.cgi?NORMAL=1, accessed on 1 January 2023) to confirm the existence of the G2-like domain.

### 2.2. Sequence Analysis

The structure of the exons and introns of the wheat *G2-like* genes was analyzed and visualized using the Gene Structure Display Server (http://www.gsds.cbi.pku.edu.cn, accessed on 1 January 2023). Conserved motifs in the identified wheat G2-like proteins were identified utilizing the MEME web-based application (https://meme-suite.org/meme/index.html, accessed on 1 January 2023) for protein sequence analysis. 

### 2.3. Phylogenetic Investigation and Categorization of Wheat G2-Like Genes

Phylogenetic trees of *G2-like* genes from wheat, rice, maize, and *Arabidopsis* were constructed using the Neighbor-Joining (NJ) method in MEGA7.0 (https://www.megasoftware.net/history, accessed on 1 January 2023). 

### 2.4. Analysis of TaG2-Like Gene Expression 

The gene expression data was obtained from the Wheat Expression Browser (http://www.wheat-expression.com/download, accessed on 1 January 2023). The data encompassed 10 tissues, namely the root, stem, spikelet, flag leaf, stamen, pistil, awn, glume, and lemma. Gene transcript abundance was quantified using transcripts per million (TPM) values. The heat map was generated using TBtools (v1.123) software [14].

### 2.5. Chromosomal Distribution and Gene Replication

*TaG2-like* genes were mapped to specific wheat chromosomes using the physical location information obtained from the wheat genome database. This mapping was visualized using Circos (http://circos.ca/, accessed on 1 January 2023) [15]. The Multiple Collinearity Scan toolkit MCScanX software (https://help.rc.ufl.edu/doc/MCScanX/, accessed on 1 January 2023) was utilized to analyze gene duplication events, employing the default parameters [16]. The syntenic maps were created using the Multiple Synteny Plot software available at https://www.tbtools.com/, accessed on 1 January 2023.

### 2.6. Plant Materials and Treatments

The wheat ‘Chinese Spring’ was selected in this study. The root, stem, leaf, lemma, stamen, pistil, and grain from different developmental stages were individually collected for RNA extraction. Three biological replicates were used for expression analysis, and three separate RNA extractions were then performed. These samples were subsequently utilized for qRT-PCR analysis.

The selected TaG2-like genes were subjected to qRT-PCR analysis to examine their expression patterns under various stress conditions. The seeds of ‘Chinese Spring’ were sterilized with 75% alcohol for 30 s and then soaked for 10 min with 20% sodium hypochlorite. Then seeds were rinsed with sterile water 5 times, 5 min each. Following the treatment, the seeds were planted in a growth chamber under conditions of a 23 °C temperature, 50% relative humidity (RH), and a light cycle of 16 h of light followed by 8 h of darkness. 

Stress treatment began when the seedling grew normally to the three-leaf stage. Treatments were conducted using Hoagland nutrient solution [17]. Control groups (CK) were only cultured with the Hoagland nutrient solution, while the stressed groups were cultured with Hoagland nutrient solution containing 100 μmol/L ABA (abscisic acid), 100 mmol/L NaCl, and 20% PEG6000, respectively. The cold stress treatment was placed in a low-temperature vernalization chamber at 4 °C for culture. The samples at 0 h, 1 h, 3 h, 6 h, 12 h, 24 h, and 48 h after stress treatment, respectively, were selected to analyze the relative expression levels of the *TaG2-like* genes in wheat leaves under different stress treatments.

### 2.7. RNA Isolation and Gene Expression Profiling

Total RNA was isolated using the Trizol protocol [18]. Real-time qRT-PCR was conducted following the manufacturer’s protocol on the CFX ConnectTM Real-Time Platform (Bio-Rad, Hercules, CA, USA). A total of 15 *TaG2-like* genes were selected to conduct real-time qRT-PCR to analyze the expression patterns of the *TaG2-like* gene family members. The primer sequences employed in this investigation are provided in Appendix A. The β-actin gene was utilized as an internal reference. The relative expressions of the *TaG2-like* gene family members were calculated using the 2^−ΔΔCT^ methods [19].

### 2.8. Determination of Subcellular Localization

The full-length gene sequences of *TaG38*, *TaG41*, and *TaG52* were amplified by PCR and subsequently cloned into the pMDC83-GFP vector. The TaG38-GFP, TaG41-GFP, TaG52-GFP, and pMDC83-GFP constructs were introduced into *Nicotiana benthamiana* leaves through *Agrobacterium*-mediated infection. After a two-day incubation period, the fluorescence of the tobacco leaves was observed using a Zeiss LSM700 confocal microscope (Zeiss, Jena, Germany).

## 3. Results

### 3.1. Identification of the G2-Like Proteins in Wheat

A total of 76 candidate G2-like proteins were obtained using HMMER3.3 and BLASTP. The protein-conserved domains of all G2-like proteins were further identified using the CDD website and SMART program. Basic information regarding the 76 wheat *G2-like* family members is presented in Appendix A. The 76 *G2-like* genes were renamed from TaG1 to TaG76 based on their order on the chromosomes.

Among the 76 G2-like proteins, two proteins (TaG14 and TaG19) were identified as the smallest protein, consisting of 354 amino acids (aa), while TaG27, TaG31, and TaG34 were the largest, spanning 684 aa, respectively. The molecular mass of the proteins varied from 24233.07 Da (TaG14) to 73659.79 Da (TaG27), respectively and the theoretical pI ranged from 5.24 (TaG60) to 9.9 (TaG35), respectively.

### 3.2. Gene Structure and Motif Composition of TaG2-Like Gene Family

According to the gene structure and motif composition of the *TaG2-like* gene family, the *TaG2-like* gene family members were clustered into three classes (Class I, Class II, and Class III, respectively), including six subfamilies (Ia, Ib, IIa, IIb, IIIa, and IIIb, respectively) (Figure 1a). Among them, Class I contained more *TaG2-like* gene family members. 

The exon–intron organization of the wheat *TaG2-like* genes showed that *TaG2-like* genes contain on average one to six exons (Figure 1b). Among them, most *TaG2-like* genes contained six exons. The motif composition of the *TaG2-like* genes was shown to be highly conservative. Typically, all *TaG2-like* genes have three same motifs (motif2, motif3, and motif8). It has been indicated that these three motifs are closely related to the conserved domain of the *TaG2-like* gene family members (Figure 1c). 

### 3.3. Phylogenetic Analysis of the G2-Like Genes between Wheat and Maize 

Among the grasses, the *G2* gene has only been identified in maize [1]. An unrooted phylogenetic tree was constructed using a total of 135 G2-like proteins from two species, including 76 wheat G2-like proteins and 59 maize G2-like proteins, respectively (Figure 2). The outcome elucidated the phylogenetic connections among the G2-like proteins. Based on the bootstrap support of the phylogenetic tree, these G2-like proteins were clustered into three classes (Class I, Class II, and Class III, respectively). The clustering results of the phylogenetic tree were very similar to the clustering results of the gene structure (Figure 1 and Figure 2), thereby being indicative of their comparable functions and evolutionary processes between maize and wheat.

### 3.4. Chromosomal Localization and Synteny Analysis of TaG2-Like Genes

The chromosomal localization showed that the *TaG2-like* genes were unevenly distributed in 19 out of 21 wheat chromosomes. No *TaG2-like* genes were detected on the homoeologous chromosomes 5B and 5D, and only one *TaG2-like* gene (*TaG36*) was located on the homoeologous chromosome 5A. Most *TaG2-like* genes were located on chromosome 7, including 7A (twelve), 7B (nine), and 7D (eight) (Figure 3a), and these genes were located on the distal ends of the chromosomes.

A total of 63 segmental copy-number variations among the *TaG2-like* genes were identified, and there were no instances of tandem duplication events observed involving the chromosomal locations of the *TaG2-like* genes (Figure 3b, Appendix A). Here, no segmental syntenic gene pairs were found on the same chromosome. Most segmental syntenic gene pairs existed among the homoeologous chromosomes, including *TaG1*, *TaG3*, *TaG5*, *TaG7*, *TaG9*, and *TaG11*, respectively. Chromosome segmental duplication events were observed not only within the same chromosome, but also between different chromosomes, such as *TaG9* on chromosome 2B, *TaG28* on chromosome 4A, *TaG9* on chromosome 2B, and *TaG33* on chromosome 4B, respectively. Additionally, several *TaG2-like* genes were found to be associated with at least three syntenic gene pair relations, such as *TaG6*, *TaG39*, and *TaG42*. These findings suggest that chromosome segmental duplication played a significant role in the evolution of *TaG2-like* genes.

### 3.5. Expression Profiling of Wheat TaG2-Like Genes in Different Tissues or Organs

The expression profiles of all 76 *TaG2-like* genes during development were analyzed using data obtained from 13 wheat organs/tissues at various developmental stages (Figure 4). There were three typical expression profiles, which were as follows: (1) *TaG2-like* genes expressed minimally, and even not expressed in all tissues during wheat development, such as *TaG4*, *TaG25*, *TaG43*, *TaG56*, *TaG57*, *TaG66*, and *TaG74*, respectively. (2) *TaG2-like* genes expressed highly in all tissues during wheat development, such as *TaG1*, *TaG27*, *TaG31*, *TaG34*, *TaG39*, *TaG40*, *TaG42*, *TaG46*, *TaG49*, and *TaG52*, respectively. These *TaG2-like* genes are likely to have fundamental and significant roles in wheat development. (3) Certain *TaG2-like* genes exhibited high expression levels specifically in certain tissues, and their expression patterns changed throughout the course of wheat development. For example, *TaG7* and *TaG26* showed high expression levels in the leaf tissue.

### 3.6. Expression Patterns of Wheat TaG2-Like Genes in Abiotic Stress

To confirm the impact of various abiotic stresses, including hormonal treatments on the expression of *TaG2-like* genes, further investigations were conducted. A total of 15 representatives *TaG2-like* members were carefully selected based on gene expression levels in various organs or tissues from 76 *TaG2-like* genes. QRT-PCR experiments were performed to investigate the expression profiles of the *TaG2-like* genes in response to different treatments (Figure 5). The findings demonstrated that the expression of the *TaG2-like* genes was significantly stimulated/suppressed by multiple treatments. For example, *TaG12* exhibited a significant response to the cold, ABA, NaCl, and PEG (polyethylene glycol) treatments. *TaG30* was induced by all the treatments examined, except for PEG stress. Conversely, several *TaG2-like* genes were concurrently induced or repressed by a single treatment. For example, three *TaG2-like* genes (*TaG7/16/30*) were induced by ABA treatment, and thirteen genes (*TaG2/6/9/10/12/17/19/20/21/23/24/25*) of *TaG2-like* were induced by ABA treatment, respectively. Interestingly, the transcript levels of most *TaG2-like* genes were down-regulated by the stress treatment. 

### 3.7. Subcellular Localization of TaG2-Like Proteins

G2-like proteins function as transcription factors in plants [2]. To investigate the subcellular localization of the G2-like proteins in wheat, we randomly selected TaG38, TaG41, and TaG52 as representatives, fused the entire coding regions (CDS) of these genes with GFP, and performed a transient expression experiment in *N. benthamiana*, revealing that the GFP protein was situated in both the nucleus and cytoplasm, while the TaG38-GFP, TaG41-GFP, and TaG52-GFP fusion proteins were found to localize in the nucleus, showing an overlap with the nuclear marker H2B-mCherry (Figure 6) [20].

## 4. Discussion

### 4.1. Characteristics and Evolution of Wheat G2-Like Gene Family Members

*G2-like* gene families have been identified in multiple species to date [3,4,7,11,12]. However, there have been few studies published on the identification of the wheat *G2-like* gene family. With the advancement in bioinformatics, the release of the wheat reference genome (Ref-Seq-v1.1) (IWGSC, 2018) was accomplished and made available, representing the current and most precise wheat chromosome-scale assembly, which promoted the study of *G2-like* gene family. For example, a previous study had identified 31 differential expressed transcriptome gene families of *G2-like* that associated with the wheat heading time [21]. In this study, all the *G2-like* gene family members were identified, and their evolution was studied. Wheat is a complex hexaploidy plant containing three sub-genomes, A, B, and D, and has 21 chromosomes [22]. The absence of *G2-like* genes on chromosome 5 (5B and 5D) could be attributed to the gene recombination or modification events involving redundant genes during the evolutionary process of the wheat plant [23,24]. The examination of gene architecture revealed that the majority of *TaG2-like* genes within the same subfamily exhibited comparable exon/intron arrangements, offering insights into the evolutionary connections among the *TaG2-like* genes [25]. Meanwhile, the expansion of the *TaG2-like* gene family arose from chromosome duplication, especially of the chromosome segmental duplication within the wheat genome [26].

### 4.2. Various Functions of the Wheat G2-Like Gene Family Members

The *G2-like* gene family belongs to a subfamily of the GARP superfamily [1], and the functions of *G2-like* genes were first identified as a transcription regulatory element in maize [2,27]. The *G2-like* genes have been exclusively recognized in photosynthetic eukaryotes, including green algae and higher plants, and the investigation indicated that the role of *G2-like* genes is linked to photosynthesis and the maturation of chloroplasts [4]. 

Moreover, *G2-like* genes are implicated in other diverse facets of plant growth and development. For example, in maize, stress-related experiments revealed that the *G2-like* genes could potentially be associated with cold and drought stress conditions [3]. Leaf senescence is a crucial physiological process in plants, and ORE1 acts as a balance between leaf senescence and maintenance by counteracting the G2-like-mediated transcription [28]. In the tomato plant, it has been confirmed that the degradation of the GOLDEN2-LIKE transcription factor is mediated by a CUL4-DDB1-based E3 ligase complex through ubiquitin conjugation [29]. Meanwhile, the researchers with *Arabidopsis thaliana* mutants found that GOLDEN2-LIKE was associated with chloroplast maturation influence ozone resilience via the modulation of stomatal behavior [30]. In this study, the results of the expression profiling of the wheat *TaG2-like* genes in different tissues or organs by RNA-seq data also confirmed this to an extent. *TaG2-like* gene family members have various functions in plant growth and development that remain for us to discover.

### 4.3. Abiotic Stresses Affect the Expressions of Wheat G2-Like Genes

The change of genes under abiotic stress is one of the important contents for gene function research. Recent findings have indicated that *G2-like* genes are involved in the heat stress response. For example, the heightened accumulation of *G2-like1* in the frost-resistant genetically modified *Brassica napus* has been found to impact the response to temperature stress [31]. Genetic variants resulting from single nucleotide polymorphisms (SNPs) are associated with gene functions related to cultivar adaptation in the tropical and temperate lines. These variants have also been linked to resistance against the cold and drought stresses [3]. In our study, *G2-like* genes in the same subfamily by phylogenetic trees showed more sequence variants. Correspondingly, most genes have similar trends under the same abiotic stress. The diversity of the sequences was found to be a significant factor contributing to the more diverse functions and the change in gene expression level.

## 5. Conclusions

In the present study, a thorough examination of the *G2-like* gene family in wheat was conducted. A total of 76 genes resembling *G2* were identified and subsequently organized into three primary groups based on their gene structure. Synteny analysis and phylogenetic comparisons of the *G2-like* genes provided valuable clues regarding the evolutionary characteristics of the wheat *G2-like* genes. *TaG2-related* genes play vital roles in the growth and development of wheat, as supported by their unique expression patterns in various tissues and their responsiveness to diverse treatments. The phylogenetic and gene expression investigations carried out in this study provide valuable insights into the functional characterization of the *TaG2-like* genes. These discoveries offer a valuable resource for enhancing our comprehension of the specific biological functions of the individual *G2-like* genes in wheat.

## Figures and Tables

**Figure 1 genes-14-01341-f001:**
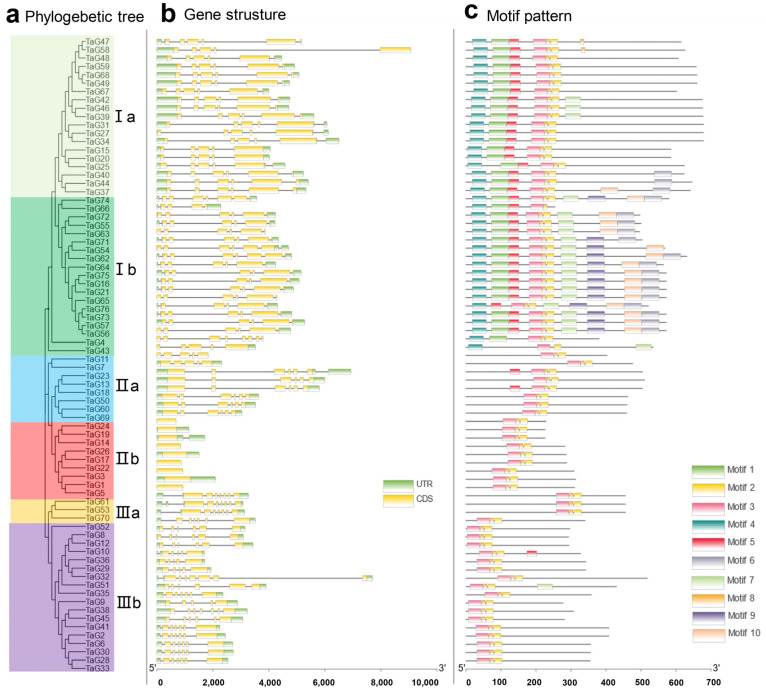
Phylogenetic relationships, gene structures, and conserved protein motif patterns of the *TaG2-like* gene family members. (**a**), The phylogenetic tree of TaG2-like proteins. Clusters are indicated with different colors; (**b**) exon–intron structures of the *TaG2-like* genes, yellow boxes indicate exons, green boxes indicate 5′- and 3′- untranslated regions, and black lines indicate introns; (**c**) the motif compositions of the *TaG2-like* genes. The motifs 1–10 are represented by the different colored boxes, with the scale at the bottom indicating the protein length.

**Figure 2 genes-14-01341-f002:**
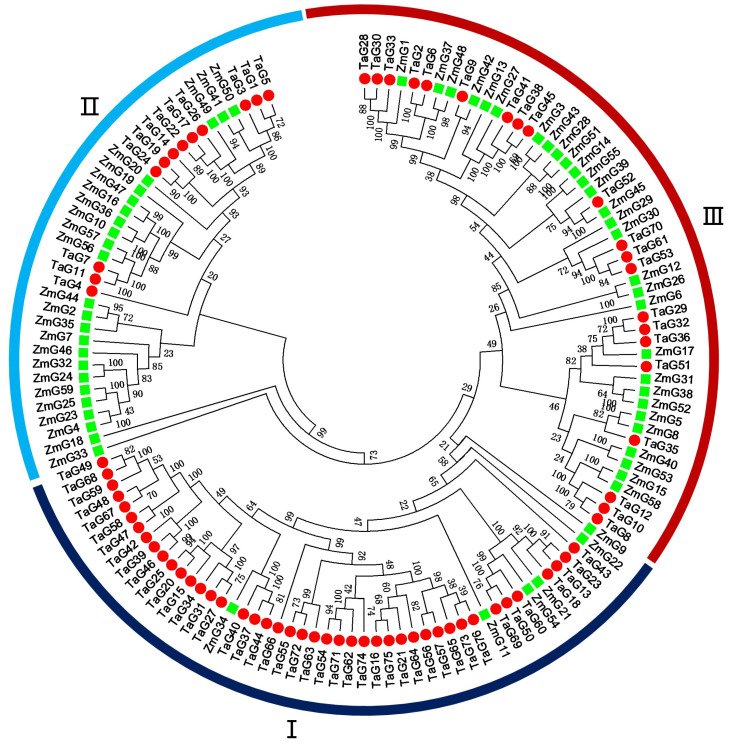
Phylogenetic tree of the G2-like proteins between wheat and maize. The red cycles represent the G2-like proteins in wheat; the green squares represent the G2-like proteins in maize. The distinct-colored arcs represent the diverse classes of the G2-like proteins.

**Figure 3 genes-14-01341-f003:**
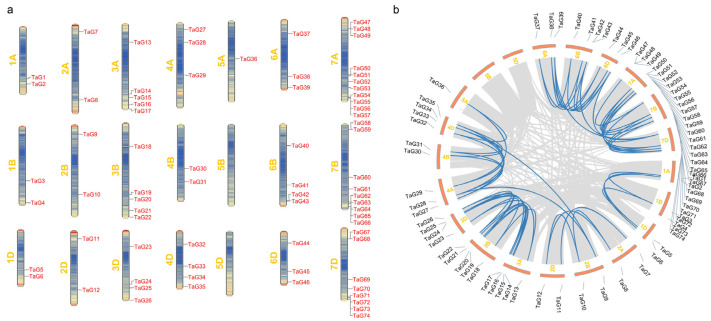
Chromosomal distribution and synteny analysis of *TaG2-like* genes. (**a**) Chromosomal distribution of *TaG2-like* genes. The gene density on each chromosome exhibits an uneven distribution, with red and blue representing the higher and lower gene densities, respectively; (**b**) an illustrative diagram presenting the chromosome mapping and interchromosomal relationships of *TaG2-like* genes. The diagrammatic representation includes gray lines representing all duplicated gene pairs in wheat, with the highlighted blue lines indicating potentially duplicated *TaG2-like* gene pairs.

**Figure 4 genes-14-01341-f004:**
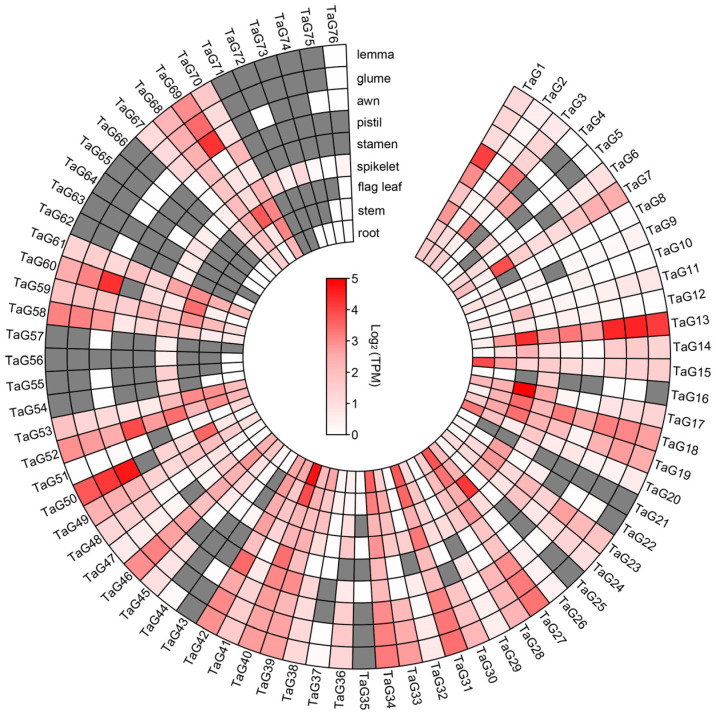
Expression profiles of the *TaG2-like* genes in various organs or tissues. The colors ranging from white to red indicate the degree of gene expression from minimal to maximal, respectively, and gray indicates that the gene is either not expressed, or that the expression level is zero.

**Figure 5 genes-14-01341-f005:**
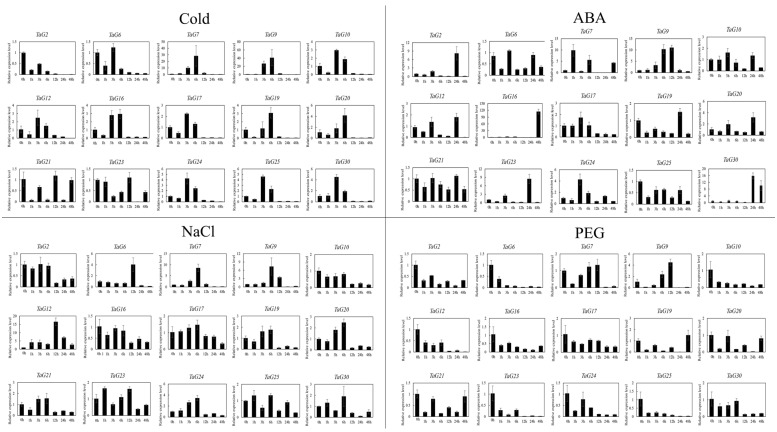
qRT-PCR results of the *TaG2-like* genes in reaction to various interventions. Information was standardized against the β-actin gene, and vertical bars represent the standard deviation. Relative expression of *TaG2-like* genes following cold, ABA, NaCl, and PEG (polyethylene glycol) treatments. Two-week-old seedlings of Chinese Spring were treated with 4 °C, 100 μmol/L ABA, 100 mmol/L NaCl, and 20% PEG-6000. Leaves were collected for expression experiments. All data are the means ± SE of three independent experiments.

**Figure 6 genes-14-01341-f006:**
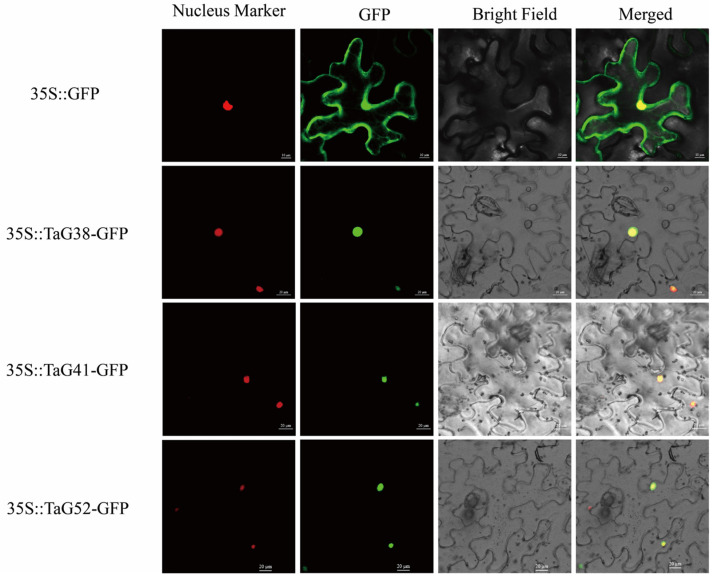
Subcellular localization results showing that the TaG2-like family numbers the TaG38-GFP, TaG41-GFP, and TaG52-GFP fusion proteins were found to localize in the nucleus. Scale bars, 20 μm.

## Data Availability

Not applicable.

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
