# Peer review of "Molecular Evolution and Genetic Variation of G2-Like Transcription Factor Genes in Wheat (Triticum aestivum L.)"

_genes, 2023, doi:10.3390/genes14071341_

Round 1
Reviewer 1 Report
1. in Figure 2, please clarify why you choose maize.
2. in figure 3, the size of the words is too small to identify.
3. in figure 6, a caption is lost in the last line.
I have no comment on the quality of the English language.
Author Response
- in Figure 2, please clarify why you choose maize.
Response 1: The repetition rate has been reduced to less than 30% according to your comment.
- in figure 3, the size of the words is too small to identify.
Response 2: Thank you very much. According to your comment, we have revised it.
- in figure 6 a caption is lost in the last line.
Response 3: Thank you very much. According to your comment, we have revised it.
Reviewer 2 Report
This paper describes G2-like transcription factors in wheat, by finding these by analysis of the publicly available Chinese Spring genome sequence and examining the expression of these and analysis of publicly available RNA expression data, and the gene expression of some of these were also determined experimentally. All this research used now standard and well-known methods.
The English is pretty good, some suggestions are below.
A point for the authors to consider: Quite some space is used in the paper to address this statement in the abstract "These genes were unevenly distributed on the 19 of 21 wheat chromosomes." My question is what is the hypothesized (presumably) "even" distribution. It is only with this, and knowing this, and the biological basis it is worked out on, can one decide on "uneven" or "even".
Some things to fix or explain:
1 in a few places "Chines Spring" is used instead of "Chinese Spring"
2 in a few places the tenses of verbs/and or verbs are non-standard; eg line 126 has "All sequences of the primers used in this study were listed in Table S1." should be "All sequences of the primers used in this study are listed in Table S1." Hopefully the journal can help with this editing.
3 line 129, is "2-△△CT methods" correct, or a symbol substitution?
4 lines 147-151, why is first sentence bold, and text smaller font size?
5 Figure 2. "The red deltas represent G2-like proteins in wheat; the green squares represent G2-like proteins in maize."; but these are red cycles.
see comments and suggestions
Author Response
1 in a few places "Chines Spring" is used instead of "Chinese Spring"
Response 1: Thank you very much. According to your comment, we have revised it.
2 in a few places the tenses of verbs/and or verbs are non-standard; eg line 126 has "All sequences of the primers used in this study were listed in Table S1." should be "All sequences of the primers used in this study are listed in Table S1." Hopefully the journal can help with this editing.
Response 2: Thank you very much. According to your comment, we have revised it. We have revised the manuscript carefully and tried to avoid any grammar or syntax error. In addition, we have asked several colleagues who are skilled authors of English language papers to check the English.
3 line 129, is "2-△△CT methods" correct, or a symbol substitution?
Response 3: Thank you very much. According to your comment, we have revised it.
4 lines 147-151 why is first sentence bold, and text smaller font size?
Response 4: Thank you very much. According to your comment, we have revised it.
5 Figure 2. "The red deltas represent G2-like proteins in wheat; the green squares represent G2-like proteins in maize." but these are red cycles.
Response 5: Thank you very much. According to your comment, we have revised it.